# The genetic identity of neighboring plants in intraspecific mixtures modulates disease susceptibility of both wheat and rice

**Rémi Pélissier**[1], **Elsa Ballini**[1], **Coline Temple**[2], **Aurélie Ducasse**[2], **Michel Colombo**[3,4], **Julien Frouin**[3], **Xiaoping Qin**[5], **Huichuan Huang**[5], **David Jacques**[3], **Fort Florian**[6], **Fréville Hélène**[3], **Violle Cyrille**[4], **Jean-Benoit Morel**[2]*

**1** PHIM, Institut Agro, INRAE, CIRAD, Univ Montpellier, Montpellier, France, **2** PHIM, INRAE, CIRAD, Institut Agro, Univ Montpellier, Montpellier, France, **3** AGAP, Univ Montpellier, CIRAD, INRAE, Institut Agro, Montpellier, France, **4** CEFE, Univ Montpellier, CNRS, EPHE, IRD, Montpellier, France, **5** Laboratory for Agro-biodiversity and Pest Control of Ministry of Education, Yunnan Agricultural University, Kunming, China, **6** CEFE, Univ Montpellier, CNRS, EPHE, IRD, Institut Agro, Montpellier, France

* jean-benoit.morel@inrae.fr

**Data Availability Statement:** All relevant data are within the paper and its Supporting Information files. The data has been deposited in a publicly available repository: "Disease levels in binary

## Abstract

Mixing crop cultivars has long been considered as a way to control epidemics at the field level and is experiencing a revival of interest in agriculture. Yet, the ability of mixing to control pests is highly variable and often unpredictable in the field. Beyond classical diversity effects such as dispersal barrier generated by genotypic diversity, several understudied processes are involved. Among them is the recently discovered neighbor-modulated susceptibility (NMS), which depicts the phenomenon that susceptibility in a given plant is affected by the presence of another healthy neighboring plant. Despite the putative tremendous importance of NMS for crop science, its occurrence and quantitative contribution to modulating susceptibility in cultivated species remains unknown. Here, in both rice and wheat inoculated in greenhouse conditions with foliar fungal pathogens considered as major threats, using more than 200 pairs of intraspecific genotype mixtures, we experimentally demonstrate the occurrence of NMS in 11% of the mixtures grown in experimental conditions that precluded any epidemics. Thus, the susceptibility of these 2 major crops results from indirect effects originating from neighboring plants. Quite remarkably, the levels of susceptibility modulated by plant–plant interactions can reach those conferred by intrinsic basal immunity. These findings open new avenues to develop more sustainable agricultural practices by engineering less susceptible crop mixtures thanks to emergent but now predictable properties of mixtures.

## Introduction

Reducing susceptibility to plant pathogens is key to maintain low pathogen burden and to control epidemics in crop fields [1]. At the plant level, susceptibility can be reduced by the action of 2 different pathways: gene-for-gene resistance, mostly based on well-known resistance

mixtures of rice and wheat", https://doi.org/10.57745/RRA3HL.

**Funding:** This work was supported by the Agence Nationale de la Recherche (ANR-16-IDEX-0006 AMUSER to CV), the CASDAR program ("BURRITOS" project to JBM), the INRAE LIA program (Plantomix project to JBM), the European Research Council (ERC) (Starting Grant Project "Ecophysiological and biophysical constraints on domestication in crop plants" ERC-StG-2014-639706-CONSTRAINTS to CV) and the National Natural Science Foundation of China (Grants 31801792 and 31960554 to HH). RP is supported by a PhD grant from Institut Agro. The funders had no role in study design, data collection and analysis, decision to publish, or preparation of the manuscript.

**Competing interests:** The authors have declared that no competing interests exist.

**Abbreviations:** DGE, direct genetic effect; IGE, indirect genetic effect; NMS, neighbor-modulated susceptibility; RST, relative susceptibility total.

genes, and basal immunity [2]. Since protection conferred by resistance genes is usually not durable, in particular in pure stands [3], plants are often left protected only by basal immunity, which reduces quantitatively the levels of susceptibility. The significant reduction of susceptibility by basal immunity is well illustrated by recent work on defense inducers that activate plant basal immunity in the fields [4]. However, finding other ways to modulate basal immunity of individual plants, in particular in a constitutive manner, would facilitate the reduction of field susceptibility levels.

One way of reducing susceptibility to pathogens at the field level consists in mixing varieties, an old practice that is experiencing a renewed interest in Europe [5–8]. For instance, septoria disease and leaf rust in wheat [9] or blast fungus in rice [10,11] can be partially controlled in varietal mixtures. Several mechanisms have been described in the literature to explain such positive diversity effects [5,12]. For instance, pathogens of one given plant genotype has a certain probability to propagate to plants to which they are not adapted, thereby leading to unsuccessful attack and a reduction of pathogen multiplication, a phenomenon known as the dilution effect. Moreover, such unsuccessful attacks can induce basal immunity that will protect plants against further infection by adapted pathogens [12,13]. Yet, meta-analyses highlight that the effects of mixing varieties on disease susceptibility are highly variable (from −40% to +40%) [14,15]. This suggests that other processes that remain understudied or even not documented are at play to modulate susceptibility.

Intraspecific plant–plant interactions can modulate individual plant susceptibility to diseases [16]. We recently identified isolated cases of varietal mixtures in rice and wheat where basal immunity and susceptibility to pathogens were modulated by healthy, intraspecific neighbor plants, a phenomenon called neighbor-modulated susceptibility (NMS; [17]). However, this study was limited to one focal plant genotype and the general response to neighbors in a variety of focal plants is unknown. Thus, the general occurrence of such phenomenon remains unknown. In addition, establishing if favorable neighbors can be found to reduce pathogen burden at the plot level would help designing varietal mixtures.

NMS mirrors other documented cases of so-called indirect genetic effects (IGEs), which characterize the ability of a genotype to modify the phenotype (here susceptibility) of another neighboring individual [18,19]. It contrasts with basal immunity that represents a direct genetic effect (DGE), which characterizes the impact of a genotype on its own phenotype (here susceptibility). Although still poorly studied in plants, IGEs have been documented on size and developmental traits [20,21]. Yet, to our knowledge, the existence of IGE on traits associated with disease resistance still needs to be demonstrated. In the perspective of using NMS to improve crop protection, the relative contributions of IGE (NMS) and DGE (i.e., susceptibility measured in the absence of neighbor) in disease limitation need to be evaluated.

Here, we measured disease susceptibility in 2 major cereal species, rice and durum wheat, in more than 200 pairs of intraspecific mixtures and their corresponding pure controls, using genetically defined varieties. Since plant breeding has mostly been conducted on pure stands and not mixtures, traits and the related genetic basis important for NMS may have been partially lost upon recent selection. Thus, to test the possible impact of modern breeding on the level and occurrence of NMS, for each study species, we selected one set of genotypes composed of elite varieties (JAPrice and ELIwheat), and one from populations that have not undergone modern selection (ACUrice and EPOwheat). Since the sets of genotypes occupy different ecological/agronomical environments, we only tested mixtures within each set of genotypes and avoided possible artifacts resulting from mixing genotypes from different sets. We used 2 major model foliar fungal pathogens of rice (*Magnaporthe oryzae*) and wheat *(Puccinia triticina)* and performed inoculations under controlled greenhouse conditions to measure disease susceptibility as a trait, in the absence of epidemics. Using a statistical model that accounts for

both IGE and DGE on disease susceptibility, we quantified the relevance of NMS and the relative contribution of neighbor effect on pathogen susceptibility in varietal mixtures.

## Results

### Broad modulation of susceptibility to pathogens in varietal mixtures in the absence of epidemics

The 201 pairs of intraspecific genotype mixtures grown in pots under controlled conditions were inoculated with fungal foliar pathogens, and disease susceptibility was monitored before any possible pathogen dispersal. Each matrix consisted in all possible pairs within each set of genotypes. Pairs of genotypes from different sets were not tested as these sets are not ecologically/agronomically compatible. We thus produced 4 matrices (JAPrice, ACUrice, ELIwheat, and EPOwheat) of susceptibility levels that were used for subsequent analyses. For each pair in a given pot, we created an index called relative susceptibility total (RST) similar to the one used to compare yield in mixed and pure cultures in the field (see Methods). RST is a relative measure of susceptibility of a mixture in a pot compared to the average values of pure stands in separate pots. At the pot level, for each of the 4 sets of intraspecific mixtures, the average of RST was significantly different from 1 (**Fig 1**), indicating that genotypes expressed different disease susceptibility depending on whether they were grown in mixture or pure conditions. Average disease susceptibility of rice to *M. oryzae* in mixtures was increased by 4% and 12% in the JAPrice and ACUrice matrices, respectively. Average susceptibility of wheat to *P. triticina* was reduced by 10% and 16%, respectively, in ELIwheat and EPOwheat matrices when grown in mixtures.

Besides this change of susceptibility at the pot level, we examined susceptibility effects at the level of individual focal plants. The distribution of the average susceptibility levels of individual plants (see Methods) was compared between the pure and the mixture conditions (**S1 Fig**). The overall susceptibility of focal plants in rice mixtures was significantly higher in the ACUrice ($p = 0.003$) but not in the JAPrice ($p = 0.64$) matrices, suggesting a tendency of interactions between genotypes to increase disease susceptibility. In wheat, the distribution of susceptibility in mixtures was significantly lower in both matrices ($p = 0.009$ for ELIwheat and $p = 0.041$ for EPOwheat).

### Specific interactions between focal and neighbor plants are the major effect explaining NMS

To investigate which of basal immunity (DGE), global neighbor effect (IGE), or specific interactions between focal and neighbor plants (DGE:IGE) contributed the most to the observed variations in susceptibility, we compared the outputs of a model designed for analyzing DGE and IGE (Model A) (see Methods; **Table 1**).

DGEs were significant in 3 out of the 4 populations tested (JAPrice, ELIwheat, and EPOwheat), indicating that intrinsic basal immunity is genetically variable in these populations. DGE accounted for up to 15% of the observed variation in susceptibility (JAPrice). In the case of the ACUrice matrix, there was no significant DGE, suggesting that the genotype of the plant had no effect on susceptibility levels to the fungal strain used. These low levels of DGE may simply result from the fact that all ACU lines come from a unique landrace. Remarkably, in 3 of the matrices tested (ACUrice, ELIwheat, and EPOwheat), the percentage of variation explained by specific focal–neighbor plant interactions reached comparable levels or was even higher (ACUrice matrix) than DGE. The IGE of neighbors on susceptibility was significant for JAPrice and EPOwheat matrices but explained only a very small proportion of the variation

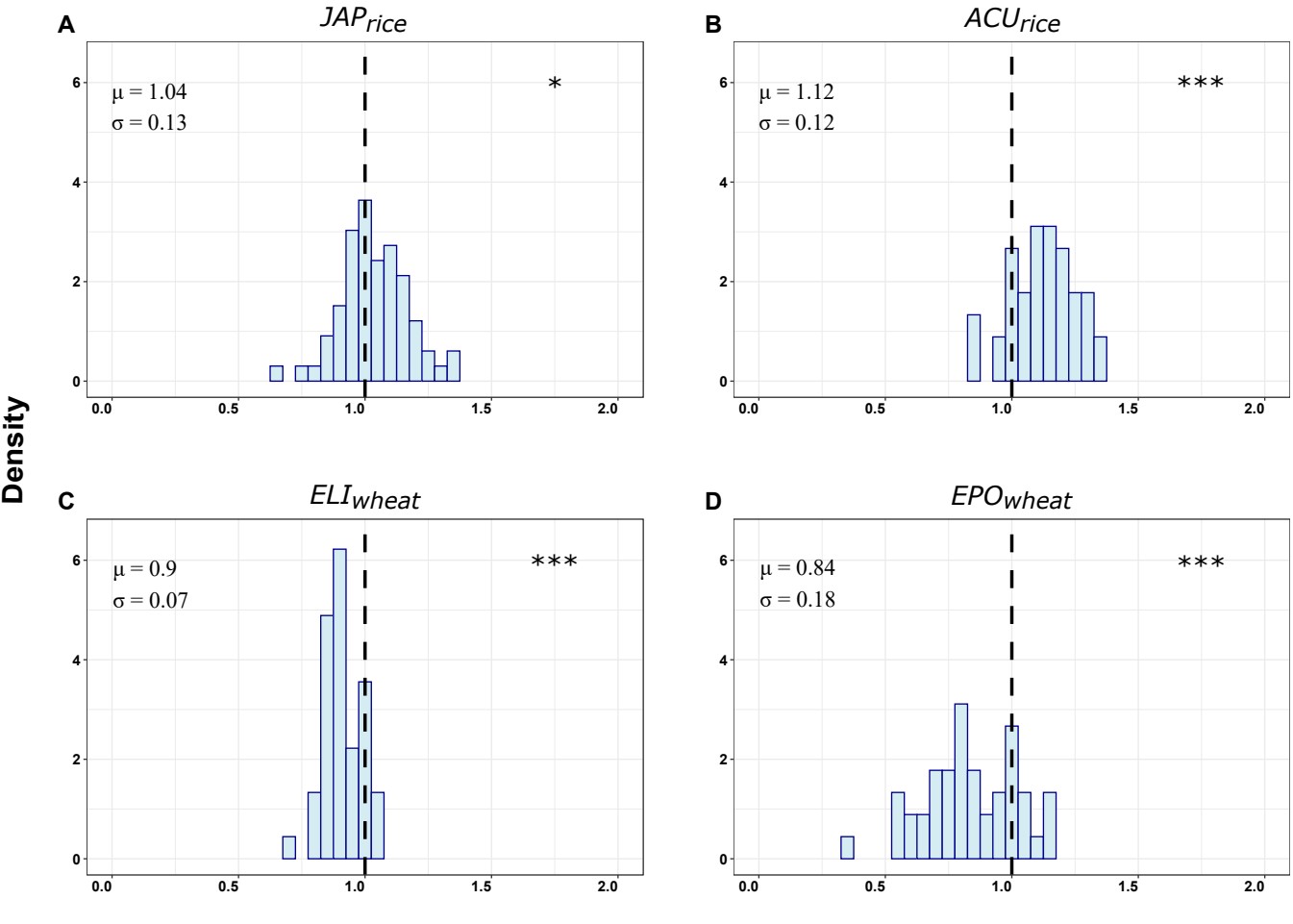

**Fig 1. Effect of genotype mixing on disease susceptibility distribution in rice and wheat populations.** Distribution of density of the relative susceptibility index (RST) for the 4 matrices of plant–plant interactions tested. At the pot level, mixing effect on disease susceptibility was quantified with the RST index, which represents the ratio between the average susceptibility of the 2 genotypes in mixture divided by the mean of the susceptibility of the 2 genotypes in pure stands (see Methods). Means (μ) and standard deviations (σ) are reported. A star symbol indicates a mean RST significantly different from 1 according to a t. test (NS: $p > 0.1$,.: $p < 0.1$; *: $p < 0.05$; **: $p < 0.01$, ***: $p < 0.001$). RST index was calculated for each pair of each of the four matrices of plant–plant interactions tested in this study: Elite temperate japonica (A: JAPrice, $n = 66$ pairs/pots), Acuce lines (B: ACUrice, $n = 45$), Elite Durum wheat (C: ELIwheat, $n = 45$) and lines from pre-breeding population (D: EPOwheat, $n = 45$). The data used can be found at https://doi.org/10.57745/RRA3HL.

(up to 1.4%) compared to DGE (8.5% to 15.3%). In contrast, the effect of specific focal–neighbor plant interactions (DGE:IGE) was significant and strong in ACUrice and EPOwheat matrices (6.9% and 7.8%, respectively). Thus, in 3 of the 4 matrices tested, the genotype of the neighbor strongly and significantly contributed to the susceptibility phenotype of the focal plant considered.

## Some genotypes are better than others in reducing susceptibility in their neighborhood

To go deeper into the identification of specific focal–neighbor plant interactions displaying NMS, we identified in the 201 pairs tested the situations in which the modulation of susceptibility of the focal plant by the presence of an intraspecific neighbor was most significant. The

**Table 1. Effect sizes of basal immunity and neighbor effects in mixtures calculated from the IGE:DGE model.**

| Set of genotypes | Species | Effect of basal immunity (DGE) | Global neighbor effect (IGE) | Specific interaction between focal and neighbors (DGE:IGE) |
|---|---|---|---|---|
| JAP$_{rice}$ (elites varieties) | **Rice** (*Oryza sativa* spp. *temperate japonica*) | 15.3%*** | 1.2%** | 4.7% |
| ACU$_{rice}$ (lines from landrace) | **Rice** (*O. sativa* spp. *indica*) | 0.6% | 0.6% | 6.9%*** |
| ELI$_{wheat}$ (elites varieties) | **Durum Wheat** (*Triticum turgidum*) | 4.4%*** | 0.3% | 5.1% |
| EPO$_{wheat}$ (inbred lines) | **Durum Wheat** (*T. turgidum*) | 8.6%*** | 1.4%** | 7.8%*** |

susceptibility of at least 1 member of the pair was significantly affected by the identity of its neighbor in 23 mix situations (11% of the cases; **Fig 2**).

In rice, the susceptibility of the focal genotype was significantly higher (up to 67% for MAR cultivated with M20) in 11 mixtures compared to its respective pure stand, whereas it was lower in 2 combinations. Out of these 13 significant genotype combinations, there was no observed case of a significant increase of susceptibility in one genotype associated with a significant reduction of susceptibility in the other. In wheat, 10 mixes corresponded to situations where the susceptibility of the focal plant was significantly affected by the identity of its neighbor. In all of them, the neighbor reduced the susceptibility of the focal plant, with 2 mixtures (line 6 with 67 and line 74 with 295) where each focal genotype of the mixture showed reduced susceptibility (from 75% to 88%) when grown with the other.

Some neighboring genotypes affected the susceptibility of several focal plants in a consistent manner: The rice variety LUX and the wheat line 68 generally increased susceptibility in their neighborhood, and, conversely, the rice variety MAR and the wheat line 74 reduced it (**Fig 2**). Because of the overrepresentation of the mix condition (11 or 9) compared to the pure control condition (1), it was not statistically appropriate to identify focal plants that were generally responsive to their neighbors. However, it seemed that genotypes like line 15 or variety MAR in rice, and line 309 or variety OBE in wheat, were affected for their susceptibility by several intraspecific neighbors. Beyond these specific significant cases of focal–neighbor interactions, when considered altogether, neighbors globally reduced the susceptibility of focal plants in wheat by 4% to 10% and increased it in rice by 9% to 16% (**S1 Fig**). Finally, there was no correlation between the level of susceptibility of the neighbor genotype and the NMS phenotype on a given focal plant (**S2 Fig**), consistent with previous results that NMS does not require the neighbor to be infected [17].

## Discussion

This study sheds lights on the general occurrence and amplitude of the recently discovered NMS phenomenon in rice and durum wheat [17]. Out of the 201 pairs tested in diverse sets of rice and wheat genotypes, we identified 23 intraspecific mixtures (approximately 11%) where disease susceptibility was modulated by plant–plant interactions. Thus, NMS is a relatively frequent phenomenon. In rice, we observed both significant positive and negative effects of the neighbors on the susceptibility of focal genotypes, suggesting that the output of plant–plant interactions can be variable. We identified some neighbor genotypes that had global effects on the susceptibility of most focal plant tested, suggesting the existence of neighbors having generic effects. Understanding how some genotypes modulate the susceptibility of their

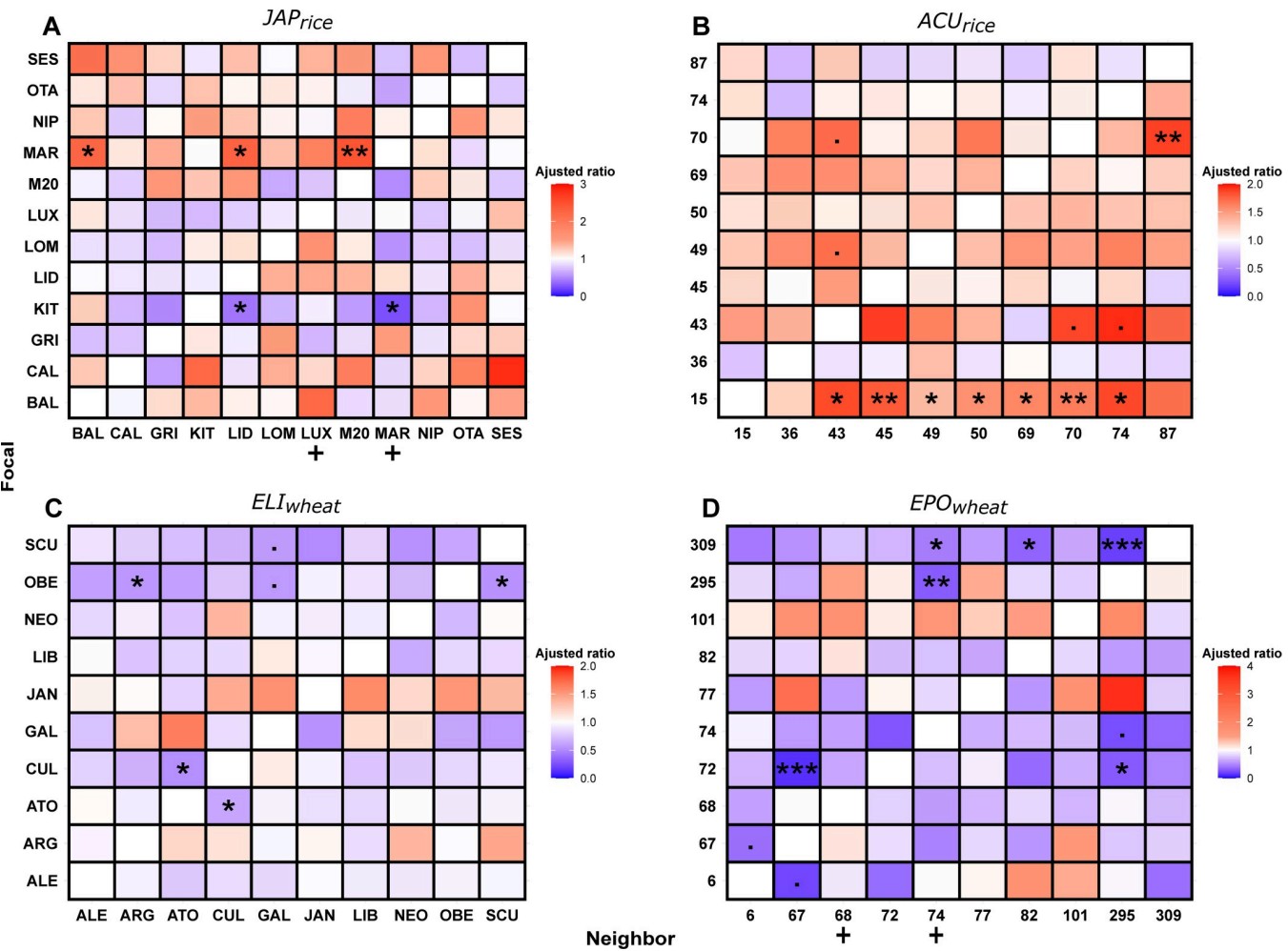

**Fig 2. Modulation of disease susceptibility by intraspecific interactions.** The susceptibility of individual focal genotypes in all possible mixtures and pure stands from Fig 1 is represented as heatmap for rice (JAPrice and ACUrice) and wheat (ELIwheat and EPOwheat). Each square corresponds to the adjusted ratio (see Methods) of the susceptibility to pathogens of a given focal plant genotype ($y$ axis) cultivated in presence of a given neighbor genotype ($x$ axis) compared to its pure stand. The value for each pure stand was thus equal to 1 and is indicated by a white color in the diagonal. A red and blue color, respectively, indicates that the focal plant becomes more and less susceptible in the presence of the considered neighbor. Adjusted ratios were used for a comparative representation of the neighbor effect on each focal plant. The figure combines the results of 2 statistical analyses performed with adapted models (described in Methods). Model A: the + symbol above the name of a neighbor plant indicates a statistically significant different group generated by a Tukey HSD test. For instance, in the JAPrice matrix, genotypes cultivated with the MAR genotype are on average more resistant than in pure stand. Model B: a * star symbol in a square indicates a statistical difference detected by a Dunnett test performed for each focal plant, with the pure stand as reference (.: $p < 0.1$; *: $p < 0.05$; **: $p < 0.01$, ***: $p < 0.001$). The data used can be found at https://doi.org/10.57745/RRA3HL.

conspecifics and what makes others responsive to their neighbors is critical to understand how plant–plant interactions affect the susceptibility to third parties that are pathogens.

The demonstration that NMS is a case of IGE was made using an appropriate model of disease susceptibility in 4 matrices representing a total of 201 pairs. Neighbor effects (IGE and DGE:IGE) significantly affected focal plant susceptibility in 3 out of 4 matrices, supporting the idea that the genetic identity of the neighbor mediates disease susceptibility to pathogens in plants in the absence of epidemics. Thus, NMS can be considered as a case of IGE, which, to our knowledge, was only observed once, in natural environment [22]. Quite remarkably, we found that in the ACUrice and EPOwheat matrices, susceptibility modulation arising from interactions between genotypes (IGE:DGE) was higher than, or similar to, susceptibility

conferred by intrinsic basal immunity (DGE). NMS could reduce susceptibility up to 88% (for EPO line 72 cultivated with line 67) or enhance it up to 67% (for MAR cultivated with M20 in the JAP matrices). Thus, the contribution of NMS to disease reduction is potentially high in intraspecific communities such as varietal mixtures.

Across all matrices, we found no evidence of a trade-off for NMS since the reduction of susceptibility in a given focal plant was not systematically associated with an increase of susceptibility of the neighbor. Among all the mechanisms that could explain NMS (presented in [16]), we cannot exclude a case of intraspecific competition for resource. In contrast, in wheat, there was a global, unilateral, and positive effect of mixtures, with focal plants showing lower disease susceptibility to the pathogen tested. This held true whatever the neighboring genotype, suggesting in this case an example of apparent cooperative behavior in plants [23].

Interestingly, IGE or DGE:IGE contributions were much less pronounced and mostly not significant in elite varieties (JAPrice and ELIwheat matrices) than in populations that have not undergone modern selection (ACUrice and EPOwheat matrices). The ACUrice matrix is composed of lines that belong to a landrace known to be composed of heterogeneous genotypes that were grown together for hundreds of years in the same paddy fields [24]. The EPOwheat matrix was made with lines that have recently been randomly selected from a large genetic basis with wheat ancestors [25]. In contrast, the JAPrice and ELIwheat matrices were made using lines that have not been grown together and come from separate breeding programs. We speculate that NMS has been maintained in coevolving genotypes and was possibly eliminated by breeding in japonica rice and durum wheat elite varieties. If this holds true, using NMS for designing varietal mixtures that display low levels of disease susceptibility may require to retrieve interesting genotypes for NMS in crop ancestors, which might have been lost during domestication and breeding like other traits [26].

Genetic distance has been invoked as a driver for the outcome of plant–plant interactions [27–30]. To test whether genetic distance between the members of each mixture influenced NMS, we quantified the correlation between RST and the genetic distance between the components of each pair (**S3 Fig**). No significant correlation was found in any of the 4 populations tested between genetic distance and RST. A locus-by-locus approach might help identify the mechanisms that trigger IGE and their interactions with DGE [31,32]. Recently, we demonstrated that NMS is amenable to genetics and identified a locus in rice neighbors that modulate resistance in their neighborhood [33].

Thus far, epidemiological interactions created by the introduction of diversity within fields were put forward to explain disease control in varietal mixtures [34,35]. In this study, disease levels of plants grown under controlled conditions, in the absence of epidemics, were strongly dependent on the identity of the neighboring genotype they grew with. Our finding raises questions on the possible occurrence of NMS in the field, with possibly antagonistic effects between NMS and epidemiological interactions resulting from diversity. For instance, recent results suggest that genetically defined plant–plant interactions could reverse most positive effects produced by diversity in the field on reducing durum wheat susceptibility to septoria disease [32]. Similarly, we observe detrimental effects of NMS in the case of rice infected by *M. oryzae*, contrasting with the observation in the field that susceptibility to *M. oryzae* is reduced by 20% to 94% in mixtures [10,11]. Thus, the removal of NMS resulting in an increase of susceptibility could result in a strong improvement of disease control. However, in the case of wheat and leaf rust, the discovery of a positive, unilateral effect of intraspecific interactions on disease reduction opens new perspectives for improving mixtures. Considering NMS as a property emerging from plant–plant interactions renews our way of studying the ecological and evolutionary drivers in intraspecific communities. From an agroecological perspective, designing efficient varietal mixtures is a major challenge [36–38]. In this context, our study

suggests that the indirect effects of plant–plant interactions on pathogen susceptibility could be used to design varietal mixtures with embedded crop protection.

## Methods

### Plant material and growth conditions

We selected in each species (rice and wheat) a set of elite varieties and a set of lines originating from highly diversified populations. We chose genotypes to maximize the variability in genetic distance between genotypes in each set. For rice, we used a set of 19,997 SNPs common to the JAPrice and the ACUrice genotypes [39]. For wheat, we used a set of approximately 46,000 SNPs common to the ELIwheat and the EPOwheat genotypes [40]. We calculated whole-genomic distance among pairs of genotypes by computing a shared allele index with DARwin software [41].

In order to reduce the strong impact of resistance genes on susceptibility phenotypes, we excluded the genotypes that were completely resistant to the pathogens used and only kept genotypes that were susceptible to the strains used. Thus, for each genotype, the susceptibility value measured is a proxy of its basal immunity. For rice, we selected 10 varieties in the *O. sativa* ssp. *temperate japonica* cultivated in Europe (Italy and France) from [39] (JAPrice), and we added the 2 reference varieties Kitaake and Nipponbare. As a second rice set, we selected 10 genotypes in the highly diversified, cultivated landrace called Acuce of *O. sativa* ssp. *indica* from the Yuanyang terraces [24] (ACUrice). For wheat, we selected a set of 10 durum wheat varieties (ELIwheat) from a durum wheat collection of 78 commercial lines produced by French private companies [40]. For the second wheat set, we used 10 inbred lines (EPOwheat) from an evolutionary pre-breeding population [25]. These cultivars are derived from a population resulting from crosses between nondomesticated wild emmer wheat, landrace, and elite germplasm. For each set, we grew each genotype in presence of itself ("Pure" condition), and binary mixtures in the presence of each other genotype belonging to the same set ("Mix" condition).

We designed 4 matrices of all possible pairs of genotypes in each set. This represented 66 pairs for the JAPrice matrix and 45 pairs of lines from each of the ACUrice, ELIwheat, and EPOwheat matrices, for a total of 201 pairs. Inoculation for susceptibility evaluation was performed on all plants in each pair. Thus, in the mix conditions, each genotype was alternatively considered as a focal and as a neighbor within the same pot; mix of genotypes A and B produced 2 phenotypic data: phenotype of A in the presence of B, and phenotype of B in the presence of A. This represented a total of 402 intraspecific mixtures and 42 corresponding pure stands. For rice, 8 plants (4 of each genotype in mixtures and 8 of the same genotype in monogenotypic pots) were grown in plastic pots (9 × 9 × 9.5 cm). For wheat, 6 plants (3 of each genotype in mixtures and 6 of the same genotype in monogenotypic plots) were grown in plastic pots (7 × 7 × 6 cm) filled with the appropriate substrate as described in [17]. Plants were grown 3 weeks and were then all inoculated with the relevant pathogens (see below). Each pot was randomly placed in the experiment and identified by its coordinates. At least 3 identical experiments were done for each matrices inoculation. Each experiments represents at least 4 replicates for each combination (4 pots at least for each focal/neighbour association).

### Pathogen inoculation and disease assessment

For rice, we selected the multivirulent *CL26* strain [42] of hemibiotrophic fungal pathogen *M. oryzae* and performed inoculations as described in [43] with a concentration of 100,000 conidia per mL. We inoculated wheat plants with a multivirulent field isolate of *P. triticina* from southern France [44] as described in [45]. We scanned the symptoms of the latest, well-

developed leaf of each focal plant per pot (3 for wheat and 4 for rice) using a resolution of 600 pixels per inch, 6 to 7 days after inoculation. Plants with retarded growth were not scored. We analyzed images with LeAFtool (Lesion Area Finding tool), a home-developed R package available on GitHub depository [46] to obtain lesion number and leaf area. Parameter values used for image analysis were at least 10,000 pixels for leaves and 50 pixels for lesion areas, with a blur at 1. To account for outliers and software mistakes, we removed from the analysis any lesion with aberrant size. Finally, we estimated leaf susceptibility by counting the number of a lesion per $cm^2$ of leaf area. The disease data (S1 Data) can be found at https://doi.org/10.57745/RRA3HL.

## Statistical analyses

a) **Relative susceptibility of the mixtures**

All statistical analyses were performed using R (www.r-project.org). To compare in Fig 1 the susceptibility of each mixture to the susceptibility of its 2 pure stand components, we calculated the RST index, inspired from the response ratio used in [15], using the following formula:

$$RST\mathbf{ij} = \frac{S\mathbf{ij} + S\mathbf{ji}}{S\mathbf{ii} + S\mathbf{jj}}$$

where *RSTij* is the RST for the mixture *ij*, and *Sij* is the LSmeans of the susceptibility (see below) of the focal genotype *i* in the presence of the genotype *j* as neighbor. Since pure stand controls (*Sii* and *Sjj*) were made of twice the same genotype, the *Sii* and *Sjj* values were produced with twice values than *Sij* and *Sji*. Under no mixing effects, RST equals 1. RST < 1 indicates that the mixture is less susceptible than the average of the 2 pure stand components, whereas RST > 1 means that the mixture is more susceptible than the average of the 2 pure stand components. For each matrix, significant difference from 1 of RST was tested using T.test.

b) **Statistical models for direct and indirect genetic effects on susceptibility**

We tested whether focal plant susceptibility responds to neighbor identity using 2 different linear models fitted with the lm function of R base. Model A accounted for the effect of the focal genotype, known as the direct genetic effect (DGE), the effect of the neighbor, known as the indirect genetic effect (IGE), and the specific interaction between the focal genotype and the neighbor genotype (DGE:IGE) on disease susceptibility of the focal genotype, as follows:

$$\sqrt{Sfocal} = Xb + I_f + I_n + I_f\,I_n + \epsilon$$

where *Sfocal* denotes the susceptibility of the focal plant expressed as the number of lesions by $cm^2$ of leaf, b is a vector of fixed effects including block and position, $I_f$ the identity of the focal plant (DGE), $I_n$ the identity of the neighbor plant (IGE), $I_f I_n$ the interaction between the focal genotype and the neighbor genotype (DGE × IGE), and $\epsilon$ the residual error. As already described in [17], square root transformation was used to correct for normality and homoscedasticity. For each of the 4 sets of genotypes, sequential type 1 ANOVA analysis was performed using the anova function of R base, then the proportion of variance explained by DGE, IGE, and DGE × IGE was computed using $\eta^2$ metric. The $\eta^2$ metric (shown as percentage in Table 1) represents the variance explained by a given variable from the remaining variance after excluding the variance explained by other factors. Model A was also used to identify neighbors with generic effects on focal plants by Tukey HSD test and calculate Least Square

means (LSmeans, using the emmeans R package) of susceptibility of each focal in mixtures. LSmeans were then used to calculate adjusted ratio for each focal in mixture, which corresponds to modulation of susceptibility by the neighbor, as follows:

$$Adjusted\ ratio\boldsymbol{ij} = \frac{S\boldsymbol{ij}}{S\boldsymbol{ii}}$$

where $S\boldsymbol{ij}$ is the LSmeans of the susceptibility of the focal genotype $\boldsymbol{i}$ in the presence of the genotype $\boldsymbol{j}$ as neighbor, and $S\boldsymbol{ii}$ the LSmeans of the susceptibility of the focal genotype $\boldsymbol{i}$ in pure stand.

To identify particular combinations of genotypes for which the susceptibility of the focal genotype was significantly affected by the identity of its neighboring genotype, we applied a second linear model (Model B) to each focal genotype. Model B was used to perform Dunnett test for each focal genotype using the pure stand as reference to identify specific neighbor genotype affecting focal susceptibility:

$$\sqrt{Sfocal} = Xb + I_n + \epsilon$$

Where $\gamma focal$ denotes the susceptibility of the focal plant expressed as the number of lesions by cm$^2$ of leaf, $\mathbf{b}$ a vector of spatial effects due to experimental design including block and position, $\boldsymbol{I_n}$ the identity of the neighbor plant (IGE), and $\epsilon$ the residual error. Square root transformation was used to correct for normality and homoscedasticity.

## Supporting information

**S1 Fig. Disease susceptibility of wheat and rice in intraspecific mixtures.** Disease susceptibility means (number of lesions / leaf cm$^2$) were quantified for each focal plant in the 2 different conditions: Pure stand (pure) in blue and mixture (mix) in red. LSmeans were calculated according to Model A described in Method. LSmeans density is represented for each set of genotypes. A: Elite temperate japonica (JAPrice) ($n$ = 132 mix and 12 pure); B: Acuce lines (ACUrice) ($n$ = 90 mix and 10 pure); C: Elite Durum wheat (ELIwheat) ($n$ = 90 mix and 10 pure); D: lines from pre-breeding population (EPOwheat) ($n$ = 90 mix and 10 pure). A star symbol indicates a statistical difference detected by ANOVA performed on the model A described in Methods between the pure and the mix conditions (NS: $p > 0.1$; *: $p < 0.05$; **: $p < 0.01$, ***: $p < 0.001$). The data used can be found at https://doi.org/10.57745/RRA3HL. (PDF)

**S2 Fig. Relation between susceptibility of the neighbor and its capacity to modulate susceptibility in a focal plant.** Modulation of disease susceptibility in focal is the fold change of susceptibility in focal plant by a neighbor genotype compared to its value in pure stand. For each set of genotypes, Elite temperate japonica (A: JAP$_{rice}$, $n$ = 132), Acuce lines (B: ACU$_{rice}$, $n$ = 90), Elite Durum wheat (C: ELI$_{wheat}$, $n$ = 90), and Pre-breeding population (D: EPO$_{wheat}$, $n$ = 90), this modulation of disease susceptibility in focal was compared to the susceptibility value of the corresponding neighbor. Pearson correlation were calculated and R$^2$ and p.value are shown. The data used can be found at https://doi.org/10.57745/RRA3HL. (PDF)

**S3 Fig. Relation between genetic distance between genotypes and RST.** RST values represent the ratio between the average susceptibility of the 2 genotypes in mixture divided by the mean of the susceptibility of the 2 component genotypes in pure stands (see Methods). For each pair of 2 genotypes, we use a shared allele index as proxy of the genetic distance (see Methods). Relation between RST and genetic distance are shown for (A) Elite temperate japonica

(JAP$_{rice}$) ($n = 66$), (B) Acuce lines (ACU$_{rice}$) ($n = 45$), (C) Elite Durum wheat (ELI$_{wheat}$) ($n = 45$), and (D) Pre-breeding population (EPO$_{wheat}$) ($n = 45$). Pearson correlation were calculated and $R^2$ and p.value are shown. The data used can be found at https://doi.org/10.57745/RRA3HL.
(PDF)

**S1 Data. The data has been deposited in a publicly available repository: "Disease levels in binary mixtures of rice and wheat" (https://doi.org/10.57745/RRA3HL).** The susceptibility data measured is given for each matrix. For each matrix are provided the Modalities (summarizes for R the modality), the Experiment (Laboratory internal reference), the focal plant on which symptoms are measured, the corresponding neighbor (plant cultivated with focal), the Species (rice or durum wheat), the Population (JAP rice, ACU rice, EPO wheat, and ELI wheat), the Pathogen used (*M. oryzae* for rice and *P. triticina* for wheat), the Lesion number (total lesion number), the Lesion/surface unit (lesion number/unit surface), the corresponding condition (Mix or pure), and, when available, the genetic distance between focal and neighbor.
(XLSX)

## Acknowledgments

We thank Andy Brousse for technical support.

## Author Contributions

**Conceptualization:** Rémi Pélissier, Elsa Ballini, Huichuan Huang, Fort Florian, Violle Cyrille, Jean-Benoit Morel.

**Data curation:** Rémi Pélissier, Coline Temple, Aurélie Ducasse, Xiaoping Qin.

**Formal analysis:** Rémi Pélissier, Elsa Ballini, Julien Frouin, Fréville Hélène, Jean-Benoit Morel.

**Funding acquisition:** Huichuan Huang, Violle Cyrille, Jean-Benoit Morel.

**Investigation:** Rémi Pélissier, Elsa Ballini, Jean-Benoit Morel.

**Methodology:** Rémi Pélissier, Aurélie Ducasse, Michel Colombo, Julien Frouin, David Jacques, Fréville Hélène.

**Project administration:** Jean-Benoit Morel.

**Resources:** Rémi Pélissier.

**Validation:** Rémi Pélissier, Jean-Benoit Morel.

**Visualization:** Rémi Pélissier.

**Writing – original draft:** Rémi Pélissier, Elsa Ballini, Jean-Benoit Morel.

**Writing – review & editing:** Rémi Pélissier, Elsa Ballini, Fort Florian, Fréville Hélène, Violle Cyrille, Jean-Benoit Morel.

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
