## [Editor Report · Decision Letter 0]

2 Sep 2022

Dear Dr. Morel, 

Thank you for submitting your manuscript entitled "Genetic identity of neighbors mediates disease susceptibility to pathogens in plant mixtures" for consideration as a Short Reports by PLOS Biology.

Your manuscript has now been evaluated by the PLOS Biology editorial staff and I am writing to let you know that we would like to send your submission out for external peer review.

Once your full submission is complete, your paper will undergo a series of checks in preparation for peer review. After your manuscript has passed the checks it will be sent out for review. To provide the metadata for your submission, please Login to Editorial Manager (https://www.editorialmanager.com/pbiology) within two working days, i.e. by Sep 04 2022 11:59PM.

Kind regards,

Paula

---

Senior Editor

PLOS Biology

---

## [Decision Letter · Decision Letter 1]

2 Dec 2022

Dear Dr. Morel,

Please allow me to first apologize for the delay in the processing of your manuscript. This delay is caused by my difficulty in recruiting reviewers for your manuscript, and is further compounded by one referee promising an overdue report but failing to deliver after long delay and multiple chases. I am sorry for this unexpected event, and I thank you for your patience while your manuscript "Genetic identity of neighbors mediates disease susceptibility to pathogens in plant mixtures" was peer-reviewed at PLOS Biology. Your manuscript has been evaluated by the PLOS Biology editors, an Academic Editor with relevant expertise, and by several independent reviewers.

As you will see in the reviewer reports, which can be found at the end of this email, although the reviewers find the work potentially interesting, they have also raised a substantial number of important concerns. Based on their specific comments and following discussion with the Academic Editor, it is clear that a substantial amount of work would be required to meet the criteria for publication in PLOS Biology. However, given our and the reviewer interest in your study, we would be open to inviting a comprehensive revision of the study that thoroughly addresses the reviewers' comments. Given the extent of revision that would be needed, we cannot make a decision about publication until we have seen the revised manuscript and your response to the reviewers' comments. Your revised manuscript would need to be seen by the reviewers again, but please note that we would not engage them unless their main concerns have been addressed. 

Having discussed the reviews with the Academic Editor, we agree with reviewer #1 that it would be better to have more mechanism, however, we consider that it is beyond the scope of a short report. We also agree with the comments of reviewer #1 that there are some inconsistencies in the data and that should be addressed with new analysis or experiments. We think that you should address both of the reviewers' concerns with further experimentation, as suggested by reviewer #1, in order for us to consider the manuscript further at PLOS Biology.

We appreciate that these requests represent a great deal of extra work, and we are willing to relax our standard revision time to allow you 6 months to revise your study. Please email us (plosbiology@plos.org) if you have any questions or concerns, or envision needing a (short) extension.

**IMPORTANT - SUBMITTING YOUR REVISION**

*Resubmission Checklist*

*Published Peer Review*

*PLOS Data Policy*

*Blot and Gel Data Policy*

Sincerely,

Paula

---

Senior Editor

PLOS Biology

REVIEWS:

Reviewer #1: Plant genomics.

Reviewer #2: Evolution and ecology.

Reviewer #1: The study investigates how plant diversity could increase mean fitness and pathogen resistance. Specifically they test so called Neighbor-Modulation of Susceptibility, where the genotype of the paired neighbor plant in the pot has an effect on the susceptibility of the focal plant in the pot.  They look at elite and unselected inbread cultivar sets of rice and wheat and compare hundreds of pairs in pots.  The study aims to look at genetic neighbor effects (direct, indirect and interaction) beyond 'classical dispersal barrier' indirect effects when mixtures are planted together in the field.  Indeed both plants in the pot should recieve equal inoculations of pathogen growing under greenhouse conditions to support this.

The genotypes are compared in a full pairwise matrix for each population and species, with mono-genotype 'controls' relatively under represented along the diagonal. The central measure if Relative Susceptibility Total index, a ratio contrasting the average susceptibility of the reciprocal pairs, to the average of the pure pots. Variation in the denominator can be high from a smaller sample and ratio statistics have large, non normal variance.  log difference, would be a more stable metric. Are the results subject to this transform, potentially better than the sqrt adjustment. 

Never the less, in table 1, only the unselected lines of rice and wheat, showed significant direct by indirect genetic interaction effects, and they were in opposite directions according to Fig1 and 2.  It is strange the ACUrice did not show direct genetic effects on susceptibility, was this significant when indirect effects were excluded? Next specific pairs were tested for neighbor effects and some lines were identified as interacting with multiple other pairs.  These should be verified in a further experiment. To go further a mapping experiment (F2, locus by line) may even be able to isolate the interacting genetic factor if this was a truly major heritable interacting factor.  

Another follow up experiment would be to contrast pairs of elite vs unselected cultivars to test the hypothesis that selection has removed this neighborhood modulation potential. It is unexpected that genetic distance among lines did not predict NMS.

I am not yet convinced "our study273 brings a proof of concept.." I would like to see alternative analysis and follow up experiments which would be a standard for PLOS Biology

Reviewer #2: In this manuscript, the authors follow up on a previous paper (Pelissier et al 2021a) in which they showed that susceptibility to disease in plants (specifically rice blast and wheat rust) depends on whether they are cultivated in pure stands or in mixtures. In short, the neighboring presence of nonself conspecifics may increase the baseline defense level of the plant, thereby decreasing its susceptibility to disease compared to a pure stand in which all plants are genetically identical. Although opposite outcomes and alternative interpretations are possible, these results shed new light on the mechanisms enabling cultivar mixtures to decrease disease prevalence as compared to pure stands. While (Pelissier et al 2021a) considered 20 pairs of plant genotypes (10 for wheat and 10 for rice), the present manuscript considers 201 pairs of plant genotypes (90 for wheat and 111 for rice). The larger size of the experiment allows the authors to go further than (Pelissier et al 2021a) in several respects. They first show that in wheat, mixing has an overall negative effect on disease severity (the number of lesions per unit area), meaning that plant-plant interactions can be beneficial to growers. By contrast, mixing has an overall positive effect on disease severity in rice, which cautions against plant-plant interactions. Using linear statistical models, the authors decompose the effects of the focal plant genotype, the genotype of its neighbor, and possible interactions between neighbor and focal genotypes, on disease severity. In particular, they show that the overall effect of interactions between plant genotypes on disease severity is far from being negligible. When plant-plant interactions decrease disease severity, their effect can be comparable to that of basal immunity. Plant-plant interactions can also increase severity though. The authors show which genotypic combinations promote or challenge the effectiveness of host mixtures.

I am a theoretical plant disease epidemiologist and I cannot judge the technical correctness of the experimental methods, including the statistical ones (which seem relatively standard). I accepted to perform this review because I was interested in the topic, and I did find the manuscript interesting and pleasant to read. The authors made an original and refreshing (not to say provocative) contribution to the field of plant disease epidemiology, showing that epidemiology (i.e., interactions between infected and healthy plants) may not be that relevant to explain the effectiveness (or lack thereof) of varietal mixtures! 

My comments below are intended to improve the readability and impact of the article on a wide readership (including theoreticians):

L69: I would recommend recalling the results and limits of the previous study (Pelissier et al, 2021) in more detail to better show the novelty of the present one.

L88-91: On first reading I thought you would mix modern and "traditional" varieties but in fact you only mix modern varieties with modern varieties and traditional varieties with traditional varieties. It is unclear why you didn't mix modern and traditional varieties: could you explain?

L102: I found it unclear on first reading which matrices you were referring to and why 4 (see my previous comment). This is because the Methods arrive after the Results. Please explain a little bit more.

L179-182: could you please explain why your statistical design did not allow you to test for overall responsiveness to neighbors?

L236: I would refer to an [apparent] cooperative behaviour since neighbor-induced reduced susceptibility may be due to intraspecific competition between plants though an elevated level of defense. I realize all this is likely explained in (Pélissier et al 2021b), but I would recommend briefly recalling the possible biological interpretations behind your statistical results. For instance, how do you interpret neighbor-induced increased susceptibility in rice? In its current version, the manuscript is a bit too focused on the statistics, leaving biological interpretations implicit.

Minor corrections/suggestions:

L62: you might consider citing Clin et al 2021.

L79-81. This sentence is redundant with L75-77.

L83: it is unclear what resistance means in this context. I would rather write "susceptibility as measured in absence of neighbor".

L119: you mean a positive (not negative) effect on susceptibility, even though the effect is negative for the grower.

L138: do you mean [genetic/specific] interactions? Otherwise, it sounds like a tautology. 

L172: focal [plant] here and elsewhere

L228: you wrote 100% line 175?

L256, 261: I would suggest writing epidemiological [interactions] instead of [barriers]. In this context, the term "barrier" has a specific meaning, which is too restrictive. Also, the term [barrier] sounds as if mixtures contained epidemiological obstacles while the reason why they are effective in some cases is not that clear (as indicated in the introduction).

L303: how many plants per pot?

L306-309: why is plastic important? What are the other pots made of?

L312: each experiment represents

L352-353: I realize that this is likely uninteresting but you should nevertheless show how making the square-root transformation corrects for normality and homoscedasticity. 

L372: why gamma and not S as before?

Fig. 1: the values on the y-axis do not seem to correspond to numbers of observations (since they can be lower than 1).

References:

Clin, P., Grognard, F., Mailleret, L., Val, F., Andrivon, D., & Hamelin, F. M. (2021). Taking advantage of pathogen diversity and immune priming to minimize disease prevalence in host mixtures: a model. Phytopathology, 111(7), 1219-1227.

Pélissier, R., Buendia, L., Brousse, A., Temple, C., Ballini, E., Fort, F., ... & Morel, J. B. (2021a). Plant neighbour-modulated susceptibility to pathogens in intraspecific mixtures. Journal of experimental botany, 72(18), 6570-6580.

Pélissier, R., Violle, C., & Morel, J. B. (2021b). Plant immunity: Good fences make good neighbors?. Current Opinion in Plant Biology, 62, 102045.

---

## [Decision Letter · Decision Letter 2]

13 Jul 2023

Dear Dr Morel,

Thank you for your patience while we considered your revised manuscript "Genetic identity of neighbors mediates disease susceptibility to pathogens in plant mixtures" for publication as a Short Reports at PLOS Biology. This revised version of your manuscript has been evaluated by the PLOS Biology editors, the Academic Editor and the original reviewers.

Based on the reviews, we are likely to accept this manuscript for publication, provided you satisfactorily address the remaining points raised by the reviewers. Please also make sure to address the following data and other policy-related requests.

1. DATA POLICY:

A) Supplementary files (e.g., excel). Please ensure that all data files are uploaded as 'Supporting Information' and are invariably referred to (in the manuscript, figure legends, and the Description field when uploading your files) using the following format verbatim: S1 Data, S2 Data, etc. Multiple panels of a single or even several figures can be included as multiple sheets in one excel file that is saved using exactly the following convention: S1_Data.xlsx (using an underscore).

B) Deposition in a publicly available repository. Please also provide the accession code or a reviewer link so that we may view your data before publication.

Regardless of the method selected, please ensure that you provide the individual numerical values that underlie the summary data displayed in the following figure panels as they are essential for readers to assess your analysis and to reproduce it: Figures 1ABCD, 2ABCD, and Supplementary Figures S1ABCD, S2ABCD, S3ABCD.

**Please also ensure that figure legends in your manuscript include information on where the underlying data can be found, and ensure your supplemental data file/s has a legend.**

2. We suggest a change in the title: "The genetic identity of neighboring plants in intra-specific mixtures mediates disease susceptibility of both wheat and rice" or "The genetic identity of neighboring plants in intra-specific mixtures modulates disease susceptibility of both wheat and rice"

We expect to receive your revised manuscript within two weeks.

*Published Peer Review History*

*Press*

Sincerely,

Paula

---

Senior Editor,

pjaureguionieva@plos.org,

PLOS Biology

Reviewer remarks:

Reviewer #1: The authors have addressed my initial concerns, particularly with the publication of another validation study

"

Answer to P3) This work was under submission at the time of the submission of the current

manuscript. We used a GWAS analysis to identify the loci in the neighbour that determine the

modification of susceptibility in focal plant Kitaake (Temperate Japonica rice). Our study

demonstrates that NMS in rice focal plants is controlled by one major locus in the genome of its

neighbour. This work is now published since January 2023 (https://doi.org/10.1111/nph.18778). A

sentence was added in the discussion: "Recently, we demonstrated that NMS is amenable to

genetics and identified a locus in rice neighbors that modulate resistance in its neighborhood (33).""

Reviewer #2: The authors have satisfactorily addressed all my comments but a minor one, see below.

L247: I suggested inserting "apparent" just before "cooperative". Apparent competition occurs when two species that have no direct negative interactions share a common predator; increasing the abundance of one species increases the predator abundance, which indirectly decreases the abundance of the other species. Here, two plant genotypes share a common pathogen; the presence of a distinct plant genotype might increase the defense level of the focal genotype, resulting, as a by-product, in a decrease in susceptibily to pathogen infection. Although the plant genotypes may have no direct positive interactions (they may not cooperate), they may have indirect positive interactions through a shared pathogen. To me the results do not necessarily suggest cooperative behaviour in plants; they may equally suggest pathogen-mediated apparent cooperation due to plant-plant competition (the opposite of cooperation!). Writing "apparent cooperation" would be more prudent.

Other comments/misprints:

L57: I would suggest replacing "Many different" with "Several"

L70: please specify "genotype" just after "plant" for clarity

L80: replace ".." with "."

L207: replace ", it" with ". It"

L271: remove "pathogen"

L373: identifY

L532: remove "for each"

---

## [Editor Report · Decision Letter 3]

7 Aug 2023

Dear Dr. Morel,

Thank you for the submission of your revised Short Reports "The genetic identity of neighboring plants in intra-specific mixtures modulates disease susceptibility of both wheat and rice" for publication in PLOS Biology. On behalf of my colleagues and the Academic Editor, Cara Haney, I am pleased to say that we can in principle accept your manuscript for publication, provided you address any remaining formatting and reporting issues. These will be detailed in an email you should receive within 2-3 business days from our colleagues in the journal operations team; no action is required from you until then. Please note that we will not be able to formally accept your manuscript and schedule it for publication until you have completed any requested changes.

PRESS

Sincerely, 

Paula

---

Senior Editor

PLOS Biology
